# Impact of Sarcopenia and Myosteatosis in Non-Cirrhotic Stages of Liver Diseases: Similarities and Differences across Aetiologies and Possible Therapeutic Strategies

**DOI:** 10.3390/biomedicines10010182

**Published:** 2022-01-16

**Authors:** Annalisa Cespiati, Marica Meroni, Rosa Lombardi, Giovanna Oberti, Paola Dongiovanni, Anna Ludovica Fracanzani

**Affiliations:** 1General Medicine and Metabolic Diseases, Fondazione IRCCS Ca’ Granda Ospedale Maggiore Policlinico, Pad. Granelli, Via F Sforza 35, 20122 Milan, Italy; annalisa.cespiati@unimi.it (A.C.); maricameroni11@gmail.com (M.M.); giovanna.oberti@unimi.it (G.O.); paola.dongiovanni@policlinico.mi.it (P.D.); anna.fracanzani@unimi.it (A.L.F.); 2Department of Pathophysiology and Transplantation, Università degli Studi di Milano, 20122 Milan, Italy

**Keywords:** sarcopenia, myosteatosis, chronic liver disease, NAFLD, alcohol liver disease, viral hepatitis, therapeutic approaches

## Abstract

Sarcopenia is defined as a loss of muscle strength, mass and function and it is a predictor of mortality. Sarcopenia is not only a geriatric disease, but it is related to several chronic conditions, including liver diseases in both its early and advanced stages. Despite the increasing number of studies exploring the role of sarcopenia in the early stages of chronic liver disease (CLD), its prevalence and the relationship between these two clinical entities are still controversial. Myosteatosis is characterized by fat accumulation in the muscles and it is related to advanced liver disease, although its role in the early stages is still under researched. Therefore, in this narrative review, we firstly aimed to evaluate the prevalence and the pathogenetic mechanisms underlying sarcopenia and myosteatosis in the early stage of CLD across different aetiologies (mainly non-alcoholic fatty liver disease, alcohol-related liver disease and viral hepatitis). Secondly, due to the increasing prevalence of sarcopenia worldwide, we aimed to revise the current and the future therapeutic approaches for the management of sarcopenia in CLD.

## 1. Introduction

The term sarcopenia was introduced for the first time in 1988 by Rosenberg et al. and it is now usually defined as a syndrome characterized by a progressive and generalized loss of skeletal muscle mass (SMM) with consequent loss of strength and function [1]. In 2018, the European Working Group on Sarcopenia in Older People (EWGSOP) provided an operating definition of sarcopenia that includes three parameters: low muscle strength measured by grip strength or chair stand test, low muscle quantity/quality evaluated through imaging techniques (dual energy X-ray absorptiometry-DXA, computed tomography-CT or magnetic resonance imaging–MRI) and low physical performance by using either gait speed, a 400 m walk, time up and a go test or short physical performance battery [1]. The assessment of SMM by MRI is considered the gold standard [2] but it is rarely used in clinical studies due to the high costs. Other imaging techniques that allow the evaluation of sarcopenia are CT and DXA, although both implicate the use of ionizing radiation [3]. Bioelectrical impedance analysis (BIA) is a non-invasive and radiation-free technique that can be exploited to evaluate the SMM. BIA is not considered in EWGSOP consensus, but it is recognized as a feasible instrument to evaluate SMM in clinical studies with large sample size [4]. The values used to define sarcopenia are widely variable, depending on imaging techniques, outcomes and a reference population with differences among ethnicity, body mass index (BMI) and associated diseases [5]. 

Sarcopenia is related to increased morbidity and mortality rates, worsening of quality of life and physical disability [6]. For many years, sarcopenia has been considered only a geriatric syndrome but, in the last few years, several studies have shown a correlation between sarcopenia and chronic diseases such as type 2 diabetes mellitus (T2DM) [7], metabolic syndrome (MetS) [8] and liver diseases [9]. Therefore, sarcopenia is considered primary when age-related and secondary when related to a disease [10]. In age-related sarcopenia, the loss of skeletal muscle is continuous and greater for men than women. The estimation of percentage of SMM lost is about 5% per 10 years. Conversely, in disease-related sarcopenia the rate of SMM loss does not have a linear trend and the percentage of muscle mass lost is greater than that observed in the age-related one, with an exponential escalation in the advanced stages of the disease [11]. Due to the increase in life expectancy and the strong association between sarcopenia and chronic diseases, in 2016 sarcopenia was recognized as an independent disease by the International Classification of Disease (ICD-10) [12]. To date, the prevalence of sarcopenia in people aged 50 years old and over ranges from 2% to 25%, according to the diagnostic tools; however, in the next thirty years its prevalence is estimated to dramatically increase, becoming a major public health issue [13].

The role of sarcopenia in cirrhosis and advanced liver disease is well defined. Indeed, sarcopenia is an independent predictor of mortality, and it is associated with a higher prevalence of portal hypertension, higher rates of infection, longer hospitalization, hepatocellular carcinoma (HCC) and worse outcomes after liver transplantation [14]. Nevertheless, subsequent studies have found that sarcopenia could also manifest in the early stages of liver disease, and it is associated with earlier fibrosis development and a worse long-term prognosis [15,16]. The association between sarcopenia and chronic liver diseases (CLD) has been deepened mainly in non-alcoholic fatty liver disease (NAFLD), but data are also accumulating on the impact of sarcopenia on other causes of liver disease, such as alcohol-related liver disease (ALD) and viral hepatitis.

Myosteatosis is defined as the fatty infiltration of muscle, both in myocytes (intramyocellular fat) and in muscle fascia (intermuscular fat). Myosteatosis is entangled with several diseases including sarcopenia [17], solid cancers and T2DM [18,19]. It is diagnosed by invasive and non-invasive techniques, such as muscle biopsy, CT, MRI and magnetic resonance spectroscopy (MRS) [20]. The majority of studies conducted on a large-scale population diagnosed myosteatosis by using CT which is based on the muscle radiation attenuation. Despite CT is not able to detect intramyocellular fat, it is considered a good technique for assessing skeletal muscle fat content [21]. 

Likewise, sarcopenia, myosteatosis is related to liver diseases, especially cirrhosis and end-stage liver diseases [22] and more recent studies described its association with NAFLD [23,24] although this relationship remains to be fully elucidated.

Therefore, the first aim of this narrative review is to evaluate the prevalence and the pathogenetic mechanisms underlying sarcopenia and myosteatosis in the early stage of CLD across different aetiologies (mainly non-alcoholic fatty liver disease, alcoholic and viral hepatitis). The second aim is to examine current and future therapeutic approaches to manage sarcopenia in liver diseases.

## 2. Research Strategy and Study Selection

We conducted a narrative review by searching peer-reviewed articles about sarcopenia and myosteatosis in non-cirrhotic stages of CLD on the PubMed database. Moreover, we also collected data about therapeutic approaches of secondary sarcopenia in the same subsets of patients.

The search time limit was before November 2021 and only English papers were considered. We included experimental and observational studies, systematic review, meta-analyses, clinical trials, editorials and commentaries reporting data on the prevalence and pathogenetic mechanisms linking sarcopenia and myosteatosis to CLD and on therapeutic approaches of sarcopenia in CLD. We excluded studies that did not meet the selection criteria, meeting abstracts, duplicate publications, studies concerning primary sarcopenia, those conducted in advanced liver disease (cirrhosis, end stage liver disease and hepatocellular carcinoma) and in human immunodeficiency virus (HIV) infected patients.

We found 415 articles, which decreased to 340 after the elimination of duplicates. Of them, according to the selection criteria, 120 papers were included in this review.

## 3. Sarcopenia in NAFLD and Metabolic Associated Comorbidities

### 3.1. Prevalence of Sarcopenia in NAFLD and Associated Factors

NAFLD is defined by an excessive lipid accumulation in hepatocytes not caused by excessive alcohol consumption [25]. The prevalence of NAFLD worldwide is around 25% [26] but it is expected to increase in the coming years, paralleling the spread of T2DM and obesity. Indeed, NAFLD is closely intertwined with several metabolic comorbidities, such as insulin-resistance (IR), T2DM, dyslipidaemia and obesity [27]. NAFLD includes an umbrella of clinical conditions ranging from simple steatosis to non-alcoholic steatohepatitis (NASH), fibrosis, cirrhosis up to HCC and it will become the first cause of liver transplantation in the next years [28].

One of the first studies that showed an inverse association between sarcopenia and incidence of NAFLD was conducted in 2013 in a Korean cohort of 1848 patients [29]. Afterwards, an overwhelming number of studies confirmed this association [30,31]. Beyond NAFLD onset, sarcopenia seems to promote NAFLD progression towards NASH and hepatic fibrosis independently of other metabolic risk factors [32,33,34,35,36], as well as to increase mortality in this category of patients [37,38,39]. 

The inverse relationship between SMM and severity of steatosis has also been confirmed in the paediatric population [40], despite the lack of association with fibrosis stages [40,41], probably as a consequence of the less severe liver damage observed in paediatric patients.

Even though the link between sarcopenia and NAFLD is established, it is, however, still unclear whether NAFLD is a consequence or a cause of sarcopenia, mainly due to scant longitudinal studies. One of the first longitudinal study was conducted in 2018 by Kim et al., who followed-up, for 7 years, a cohort of 12,624 subjects from the general population and found that 15% of the cohort developed NAFLD, but the lower the tertile of skeletal muscle mass evaluated by BIA, the higher the incidence of NAFLD was, independently of major confounding factors. Moreover, among patients affected by NAFLD at baseline, subjects with the higher tertile of SMM showed a resolution of NAFLD, compared to subjects with lower muscle mass [42]. This study had several limitations, such as the use of the hepatic steatosis index (HSI) to define steatosis and the lack of validated SSM cut-off to define sarcopenia. Another interesting study conducted on 1595 Korean patients showed that baseline NAFLD was an independent risk factor for the development of sarcopenia diagnosed by DXA over a 2-year period [43], in contrast to a previous Korean study in which the age-related reduction of SMM was associated with NAFLD [44]. A recent preclinical study conducted in obese mice fed with high fat or high fructose diet displayed that sarcopenia, evidenced by a low muscle strength, was associated only with fibrosing NASH and not simple steatosis [45]. Therefore, to date, it remains unclear whether NAFLD is a consequence or rather a cause of sarcopenia.

In NAFLD patients, a condition named sarcopenic obesity is frequently observed, which is characterized by the co-presence of both sarcopenia and obesity. Sarcopenic obesity is a consequence of reduced muscle mass and increased body fat [46] and is related to higher morbidity and mortality, also when it is compared to sarcopenia and obesity alone [47]. Indeed, in NAFLD patients, sarcopenic obesity is related to an increase in liver fibrosis and NASH [36,48] and it seems to be related to HCC and its recurrence after surgical resection [49]. 

Data obtained by studies which have assessed the association between sarcopenia and NAFLD have several caveats, the most important relying on the operational definition of SMM. Indeed, if on the one hand both the EWGSOP and the Asian Working Group for sarcopenia (AWGS) [50] considered the SMM adjusted by height squared (hSMI), on the other hand the US-based Foundation for the National Institute of Health Sarcopenia Project recommended the adjustment of SMM by BMI [51]. As a consequence, studies conducted in NAFLD patients that used SMM adjusted by weight (wSMI) found an association between sarcopenia and NAFLD, but this association was not confirmed in studies that adopted hSMI as an adjusting factor [52]. However, since NAFLD is strictly correlated with body weight and BMI, by adjusting the target variable for weight could insert a bias in the analysis. Moreover, wSMI correlates to a lesser extent with muscular function than hSMI and shows a negative correlation with waist circumference, BMI and visceral adiposity [53]. Another limitation is represented by the assessment of sarcopenia, which is diagnosed only by the reduction in SMM, although the EWGSOP definition suggests also including an evaluation of muscle strength and function. Finally, the roles of the ethnic background, gender and genetic predisposition associated with the development of both NAFLD and sarcopenia have not been defined yet. Most of the studies have been performed in Asian subjects, whereas those performed in Caucasians described an association between sarcopenia and NAFLD mainly in women [34,54]. Concerning non-Asian and non-Caucasian ethnic groups only a few studies, mainly in children, investigated the relationship between sarcopenia and NAFLD. The studies confirmed the association among sarcopenia and steatosis but not with hepatic fibrosis [41].

### 3.2. Pathogenic Mechanisms of NAFLD-Related Sarcopenia

NAFLD and sarcopenia share common underlying pathogenic mechanisms, such as IR, chronic inflammation, mitochondrial dysfunction, nutritional deficiencies and reduction in physical activity [55]. Insulin exerts its action on both skeletal muscles and the liver. In muscle tissue, insulin promotes the storage of postprandial glucose through the phosphorylation of several target proteins involved in glycogen synthesis and in the translocation of glucose transporter type 4 (GLUT4) on the plasma membrane [56]. Moreover, insulin promotes muscle mass enlargement through the activation of mammalian target of rapamycin (mTOR), thus inducing the expression of muscle proteins [57] and suppressing proteins catabolism [58]. In hepatocytes, insulin plays a key role in glucose, lipid and protein metabolisms. During feeding, insulin binding to its receptor activates protein kinase B (Akt) signalling. Akt in turn promotes the phosphorylation of the transcription factor forkhead box 01 (FOXO1) with consequent inhibition of gluconeogenesis [59]. Insulin also modulates sterol regulatory element-binding protein-1c (SREBP-1c), the primary transcription factor involved in de novo lipogenesis [60]. The Akt-mTOR pathway is also implicated in protein metabolism, favouring their synthesis and reducing in turn their catabolism [61]. In the presence of IR, a reduction in protein synthesis and an increase in protein catabolism occur in myocytes [62]. These events lead to SMM reduction and to the onset of sarcopenia. Muscle mass shortening hesitates the harmful consequences of IR thus contributing to glucose intolerance and promoting gluconeogenesis, proteolysis and muscle depletion in a vicious cycle [63]. IR-driven hyperinsulinaemia upregulates hepatic SREBP-1c and reduces the β-oxidation. As a consequence, free fatty acids (FFA) and triglycerides accumulate themselves into the liver and induce steatosis [64,65].

Another target tissue of insulin is adipose tissue. In adipocyte, insulin promotes the translocation of GLUT4 on the plasma membrane, with a consequent increase in glucose uptake and the promotion of lipogenesis. In the presence of IR, lipolysis is not suppressed, thus fostering a greater release of FFAs into the bloodstream reaching liver and muscles. The exaggerated uptake of FFAs in these tissues initiates hepatic steatosis and sarcopenia [66].

Moreover, both sarcopenia and NAFLD are characterized by a chronic inflammatory state shaped by the over secretion of pro-inflammatory cytokines, such as tumour necrosis factor alpha (TNFα) and interleukin 1 (IL1) [30]. The latter, together with an increased oxidative stress, promotes protein catabolism in skeletal muscle, dampening in the amount of muscle mass [67,68]. In turn, TNFα promotes the transcription of several genes involved in lipogenesis thus enhancing hepatic fat accumulation [69]. Moreover, TNFα, through the activation of nuclear factor-kB (NFkB), perpetuates hepatic inflammation, in a vicious circle that amplifies the inflammatory milieu in both sarcopenia and NAFLD [70].

Nonetheless, even adipose tissue is targeted by TNFα and other pro-inflammatory cytokines, where they foster the development of IR [71]. In turn, the adipose tissue itself releases several pro-inflammatory cytokines (referred to as adipokines) that recruit monocytes/macrophages, which infiltrate the adipose and muscle tissues, exacerbating the inflammatory pathways and muscle atrophy [72]. Conversely, two cytokines produced by muscles and protective towards NAFLD development—namely Interleukin 6 (IL6) [73] and Irisin [73]—seem to be reduced in sarcopenia. Moreover, in a recent study enrolling 370 patients with biopsy-proven NAFLD, the authors found an association between sarcopenia, advanced fibrosis and *fibronectin type III domain-containing protein 5* (*FNDC5*) variant, encoding for irisin [74]. 

Another pathogenic mechanism shared by sarcopenia and NAFLD is represented by mitochondrial dysfunction. During aging, a reduction in mitochondrial function occurs, with a loss of protein homeostasis in skeletal muscle. Thus, damaged proteins are gathered into the muscles sustaining oxidative stress and reactive oxygen species (ROS) production, that worsens mitochondrial dysfunction and favours sarcopenia [75], especially age-related. Another mechanism involved in age-related mitochondrial dysfunction is a reduction of AMP-activating protein kinase (AMPK), a protein kinase that regulates autophagy and mitophagy. The deregulation of AMPK is involved in increased mitochondrial size and reduction of mitochondrial DNA that leads to organelles’ damage [76]. *Autophagy related protein 7* (*Atg7*) is a gene involved in autophagy and mitophagy, two processes involved in the maintenance of skeletal muscle mass. *Atg7-/-* in mice increases protein carbonylation with a consequent enlargement and swallowing of muscle mitochondria, similar to mitochondrial alterations seen in age-related sarcopenia [77]. In turn, mitochondrial dysfunction seems to be related to a reduction of GLUT4 translocation through the muscle cell plasmatic membrane, a mechanism involved in IR [78]. Increased mitochondrial mass and biogenesis in the liver has been described in NAFLD and evidence points towards inadequate mitochondrial adaptation as a central player in the progression to NASH [79]. As the intra-hepatic levels of lipid increase, there is a concomitant enhancement of the mitochondrial respiratory chain activity [80]. This compensatory mechanism prevents the progression of liver damage but, when the oxidative stress overcomes the antioxidant defence of the liver, a reduction in mitochondrial function occurs thus fostering the shift towards more severe forms of liver disease [80].

Finally, vitamin D is a well-known factor implicated in the homeostasis of skeletal muscle [81]. The active form of vitamin D binds the nuclear receptor, the vitamin D receptor (VDR), which is expressed in skeletal muscle cells and modulates the expression of several genes involved in cell regulation and proliferation [82]. Ageing is linked to vitamin D deficiency and to a lower VDR expression, together with a reduced proliferation and differentiation of myoblasts and the consequent development of sarcopenia [83]. In hepatocytes, vitamin D reduces the proliferation of fibroblasts and the production of collagen, thus protecting against liver fibrosis [84]. Moreover, vitamin D reduces lipogenesis through the downregulation of SREBP-1c. Conversely, it increases fatty acid oxidation and this effect is mediated by the activation of peroxisome proliferator-activated receptor α (PPAR-α) [85]. A deficit of vitamin D is related to IR, oxidative stress and chronic inflammation and the majority of studies conducted in both animals and humans showed an association with hepatic steatosis, independently of BMI, T2DM and MetS [86,87].

## 4. Sarcopenia in Alcohol-Related Liver Disease

### 4.1. Prevalence of Sarcopenia in ALD and Associated Factors

Likewise, NAFLD, ALD is also a public health issue worldwide with severe consequences for morbidity and mortality [88]. Alcohol exerts well characterized deleterious effects on the liver, leading to the alcoholic steatosis, steatohepatitis, and to the progression to cirrhosis and end stage liver disease [89]. Alcohol consumption is associated with sarcopenia and, in ALD, it tends to interact with liver damage to worsen the sarcopenic condition [90]. Alcohol itself represents a risk factor for sarcopenia only in the presence of liver disease, as shown in a meta-analysis, which included 13 studies thus encompassing 13,155 subjects. The authors, despite the different methods for quantifying alcohol consumption across studies, showed that alcohol did not contribute to sarcopenia development [91]. Likewise NAFLD, also in ALD sarcopenia develops in the early stage of liver disease and, despite the heterogeneity between studies, about 60% of patients with ALD are sarcopenic [92]. The presence of sarcopenia in ALD is associated with the worst clinical outcomes, such as a more severe form of alcohol hepatitis, a higher risk of pneumonia, sepsis, hepatic encephalopathy and longer hospital stays compared to patients without sarcopenia [93].

Another complication of alcohol abuse is malnutrition, a condition strictly connected to sarcopenia, especially in ALD [94]. Alcohol has a caloric value greater than protein and carbohydrate, but it does not provide vitamins and nutrients. As a result, excessive alcohol consumption satisfies the daily caloric requirements but leads to malnutrition [95]. In alcoholic hepatitis, the presence of mild or severe malnutrition is related to a one-year mortality of 14% and 76%, respectively, as shown in a study conducted on 363 subjects admitted to the Emergency Department for acute alcoholic hepatitis [96]. Since sarcopenia frequently represents the clinical phenotype of malnutrition in ALD [94], we can assume that sarcopenia is related to an increase of mortality rates in alcoholic hepatitis.

In addition to the limitations previously identified in NAFLD, the measurement of skeletal muscle in ALD is further made difficult due to fluid retention, especially when BIA is used [97]. Moreover, the majority of studies are conducted in patients with alcohol related cirrhosis or alcoholic hepatitis, while few data are available in patients with early-stage ALD.

### 4.2. Pathogenic Mechanisms of Alcohol-Related Sarcopenia

Ethanol exerts a direct damaging effect both in muscle and liver. Ethanol is metabolized in hepatocytes by a two-step oxidation process. The first oxidation is promoted by alcohol dehydrogenase and produces acetaldehyde in the cytosol. The same process is also catalysed by cytochrome P450 2E1 (CYP2E1), a member of the microsomal ethanol oxidizing system (MEOS) and by catalase in peroxisomes. MEOS and catalase act when the body needs to process a larger amount of alcohol [98]. The second process of oxidation is catalysed by aldehyde dehydrogenases in mitochondrial matrix and produces acetate [99]. Acetaldehyde is a small and neutral compound that passes the mitochondrial membrane without carriers or channels. In mitochondria, acetaldehyde interferes with hepatic ureagenesis with a consequent increase in ammonia levels [100]. Hyperammonaemia has a pleiotropic effect on muscle cells as it inhibits the mTOR complex (mTORC1), resulting in autophagy, blunted protein synthesis and sarcopenia [101]. Ammonia *per se* stimulates the synthesis of myostatin, a member of the transforming growth factor beta (TGFβ) superfamily, synthesized and expressed in skeletal muscle. Myostatin plays a negative role in myogenesis and high levels of myostatin have been observed during muscle wasting. Recently, myostatin has been negatively involved in glucose and lipid metabolism, favouring the development of IR, T2DM and obesity [102,103]. The oxidation of ethanol to acetaldehyde is also associated with the oxidative stress and production of ROS, with an increase in lipid peroxidation [104], mitochondrial dysfunction, reduction in adenosine triphosphate (ATP) production and the promotion of autophagy [105]. These pathologic events are evidenced both in hepatic and muscle cells, possibly explained by the myocytes’ expression of CYP2E1. Several cytokines, such as IL-6, IL-10 and TNFα [106], are involved in the pathogenesis of ALD similarly to NAFLD [30]. As mentioned before, pro-inflammatory cytokine secretion promotes protein catabolism in muscle cells, predisposing to sarcopenia [66,107]. Additionally, ethanol exerts both a direct and indirect profibrotic effect on the liver. As for the first, acetaldehyde promotes the transition of hepatic stellate cells (HSCs) to myofibroblasts producing an extracellular matrix (ECM). Initially, the ECM accumulates into Disse space and around terminal hepatic veins. Subsequently, collagen deposition spread throughout the liver [108]. The dysbiosis and the increase intestinal permeability due to alcohol intake [109,110] lead to an increase in circulating lipopolysaccharide (LPS) levels [111]. Pro-inflammatory cytokines and LPS, through the binding on CD14 receptors expressed in HSCs, exert a profibrotic effect via the activation of NFkB and the signal transducer and activator of transcription 6 (STAT6) [112,113]. These signalling pathways are involved in fibrogenesis, similarly to the direct effect exerted by acetaldehyde [114]. As for the indirect profibrotic effect, the hepatocellular injury promoted by acetaldehyde produces hedgehog ligands which favour ECM deposition, through the activation of several hedgehog responsive genes such as vimentin, α-smooth muscle actin (α-SMA) and GLI family zinc finger 2 [115]. A preclinical model of male rats fed with an ethanol-containing diet highlighted an increase of fibrotic tissue in skeletal muscle after ethanol assumption due to the activation of profibrotic genes [116]; however, the impact of ethanol on muscle fibrosis is poorly investigated in humans. 

## 5. Sarcopenia in Chronic Viral Hepatitis

### 5.1. Prevalence of Sarcopenia in Viral Hepatitis

Sarcopenia is also observed in patients affected by chronic hepatitis B (CHB) and C (CHC). Like NAFLD and ALD, sarcopenia is detected in all stages of liver damage, with an increase in prevalence from the non-cirrhotic stages (7.1%) to compensated cirrhosis (11.8%) and decompensated cirrhosis (21.9%) [117]. The presence of sarcopenia in CHB and CHC has a bad prognostic value, increasing the morbidity and mortality rate due to chronic infections [118]. In CHC, sarcopenia seems to precede cirrhosis and it is related to malnutrition and male sex [119]. The effect of viral eradication on sarcopenia is not well defined. A study conducted in 2020 by Endo et al. showed that the skeletal muscle index (SMI), evaluated by CT, did not improve after therapy with direct-antiviral agents (DAAs). Nevertheless, these authors found that viral eradication by DAAs stops skeletal muscle loss, in both the young and the elderly [120]. 

Few clinical studies explored the prevalence of sarcopenia and the impact of antiviral therapy on muscle mass and function in non-cirrhotic CHB patients. The first study was performed in 506 Korean CHB patients and sarcopenia was associated with advanced liver fibrosis, detected by a non-invasive score. In a subgroup of CHB patients with at least one metabolic comorbidity, such as IR, obesity, MetS or steatosis, sarcopenia was correlated with an increased risk of significant liver fibrosis (odds ratio ranging from 2.37 to 3.57) [121]. In the same cohort, the authors showed that physical activity reduces the risk of severe fibrosis in sarcopenic patients [121]. Concerning the effect of antiviral therapy on sarcopenia, Kyung et al. described a reduction in appendicular skeletal muscle mass, evaluated by BIA, during antiviral therapy in older and male patients with higher BMI and higher aspartate aminotransferase (AST) levels [122].

### 5.2. Pathogenic Mechanisms of Viral Infections-Related Sarcopenia

The pathogenetic mechanisms that link sarcopenia and chronic viral hepatitis are unclear. In chronic liver disease, sarcopenia is the result of an imbalance between protein synthesis and catabolism supported by several mechanisms, partly shared with NAFLD and ALD. Systemic chronic inflammation characterizes both CHC and CHB patients with a consequent increase in pro-inflammatory cytokines [67,123,124]. TNFα and IL1 stimulate ROS production that, acting as second messengers, activate the transcription factor NFkB. As a consequence, a higher transcription of genes encoding pro-inflammatory cytokines and ECM proteins is observed, favouring the maintenance of the inflammatory milieu [125]. As mentioned before, chronic inflammation promotes protein catabolism in muscle cells and the development of sarcopenia, probably due to the inhibition of Akt/mTOR pathway and to the induction of ubiquitin-proteasome system (UPS) [67,126]. UPS is one of the main mechanisms involved in protein degradation in muscle cells and it is activated by FOXO and NFkB [125]. Pro-inflammatory cytokines activate NFkB and inhibit Akt (that normally blocks FOXO), promoting UPS and favouring protein catabolism [127]. 

In CHC, after viral eradication, IL6 and TNFα levels are reduced, probably secondary to the reduction in systemic inflammation. This effect has been observed after therapy with both pegylated-interferon plus ribavirin [128] and DAAs, even if the improvement in pro-inflammatory cytokines is less sustained in advanced liver disease [129]. 

Moreover, it has been widely demonstrated that HCV eradication ameliorates liver fibrosis [130,131], possibly explaining the raising of serum albumin concentrations after viral clearance [132]. Elevated albumin levels may reflect the improvement in protein synthesis, rescuing the loss of skeletal muscle. An improvement in the quality of life of patients after viral eradication could encourage them to follow a balanced diet and a physical activity program, leading to muscle mass expansion [120].

A schematic overview of the mechanisms underlying sarcopenia development in the context of NAFLD, ALD and viral hepatitis is represented in Figure 1.

## 6. Role of Myosteatosis in Chronic Liver Diseases

Myosteatosis is defined as high muscle fat infiltration evaluated by imaging techniques showing a reduction in skeletal muscle density (CT or MRI). Myosteatosis is related to an increase in all causes and the cardiovascular (CV) mortality rate in the elderly, especially in men [133]. In the last few years, several studies have focused on myosteatosis in liver diseases, however mainly in cirrhotic patients or in subjects after liver transplantation. In advanced liver disease, myosteatosis acts as a prognostic factor for morbidity, mortality and adverse perioperative outcomes [22,134]. Research aiming to assess myosteatosis in the early stages of liver diseases are limited by the use of CT or MRI to measure muscle fat content and data regarding myosteatosis and CLD are still controversial. However, a linear relationship between myosteatosis and NAFLD severity has been identified [23,135], as well as in CHC patients after viral eradication, probably due to an increase in BMI [120]. 

Similar findings have been detected in metabolic-dysfunction associated fatty liver disease (MAFLD), a condition in which liver steatosis is associated with the presence of, at least, one of the criteria between T2DM, overweight or metabolic dysregulation [136,137]. In MAFLD patients, myosteatosis was associated with liver stiffness evaluated by Fibroscan, as confirmed in a large retrospective study conducted in obese patients [138].

Data in humans are sustained also by pre-clinical studies. In NAFLD obese mice, myosteatosis but not sarcopenia was associated with NASH and fibrosis independently of visceral fat or IR, possibly representing a surrogate diagnostic marker for NASH (AUROC 0.96, *P* 60; 0.0001) [44].

Several mechanisms are shared by myosteatosis and CLD, especially NAFLD, in which more experimental findings are available. Firstly, fat load in hepatocytes and muscle cells is directly related to IR, both in diabetic, obese and lean patients [139]. Fat accumulation into skeletal muscles is closely linked to visceral adipose tissue and it is a good predictor of IR [140]. The mechanism by which myosteatosis promotes IR is not fully elucidated. Probably, the intense FFA flux to muscle cells resulting from an enhanced adipose tissue lipolysis leads to IR [141]. Myosteatosis is characterized by a higher percentage of macrophage into muscle, which secretes pro-inflammatory cytokines that foster IR, ECM deposition and fibrosis development in muscle [142]. 

Secondly, an increase in cathepsin D levels has been described in both myosteatosis and NAFLD. Cathepsin D is a lysosomal enzyme associated with inflammation, lipid metabolism and NAFLD severity in in vitro and in in vivo models [143]. Cathepsin D levels seem to also be boosted in myosteatosis, probably due to higher intracellular calcium concentrations fostered by lipid accumulation in cells. Calcium overload favours the fusion between lysosomes and plasma membrane, with the release of cathepsin D and other enzymes [144]. Cathepsin D could induce lipid accumulation into cells, with consequent myosteatosis onset [145]. A recent study conducted by Nachit et al. showed an enhanced prevalence of myosteatosis in obese NASH patients and suggested that myosteatosis could reflect the histological features of NASH [24]. Conversely, they reported a lower prevalence of sarcopenia in obese NAFLD patients with a direct relationship between muscle mass and severity of liver disease, in contrast to previous literature [24]. 

The lack of large, controlled trials in humans regarding the therapeutic approaches to myosteatosis is the major limit to defining a possible treatment. Physical activity and a low caloric diet have showed a beneficial effect not only on muscle mass but also on muscle fat accumulation [146]. In a preclinical model of obese mice, the supplementation with vitamin D (7 μg/kg three times/weeks) improved the muscle IR and inflammation, but a direct effect of vitamin D on myosteatosis was not further demonstrated [147]. 

## 7. Potential Therapeutic Approaches to Sarcopenia in Chronic Liver Diseases: From Current Therapies to Novel Molecular Targets

Sarcopenia is often linked to malnutrition, and it is particularly noticeable in CLD [148]. In this association are involved metabolic and hormonal alterations, drugs and pro-inflammatory status, as well as the typical featuring of aging namely impaired gut motility, reduction in olfaction and taste, psychological impairment, difficulty with mastication [149]. Due to the importance of sarcopenia in the early stages of CLD and the increased evidence about the pathophysiological mechanisms linking sarcopenia and CLD, several therapeutic strategies have been considered to prevent and treat muscle decline.

### 7.1. Current Therapeutic Options

To date, a controlled diet and physical activity remain the suggested approaches to prevent the development and progression of sarcopenia, especially in CLD [150].

Amino acids provided by diet stimulated muscle protein synthesis and essential amino acids exert a higher anabolic effect compared to non-essential ones [151]. Isoleucine, leucine and valine are also branched-chain amino acids (BCAAs), which with oral administration are able to reduce the progression of liver disease [152]. BCAAs are metabolized in muscles and regulate protein metabolism through mTOR signalling [153]. Essential amino acids and BCAAs also have an effect on mitochondria homeostasis, fostering its biogenesis and the production of antioxidant molecules. As a consequence, diet supplementation with essential amino acids and a BCAAs mixture seems to have a greater effect on sarcopenia [154].

A high and chronic intake of added sugars, such as fructose, is related to NAFLD development and to sarcopenia. Indeed, fructose fosters the onset of IR, oxidative stress and mitochondrial dysfunction with consequent fat accumulation in the liver and muscle and increased autophagy of muscle cells. Therefore, a diet low in added sugars induces benefits by preventing hepatic and muscular damage [155]. 

As mentioned before, vitamin D is a key factor in the maintenance of muscle tissue homeostasis. In old people, daily supplementation of 700–1000 IU of vitamin D seems to reduce the frailty and the risk of falls [156]. 

Selenium, carotenoids, tocopherols, flavonoids and polyphenols are antioxidants’ agents that reduce the production of ROS and the oxidative damage. The antioxidants’ effect promotes protein synthesis and reduces protein catabolism, with improvement in sarcopenia [157].

The role of polyunsaturated fatty acid (PUFAs) is still controversial. PUFAs encompass n-3 PUFA and 6-n PUFA, both essentials as humans cannot synthetize them [158]. n-3 PUFA as docosahexaenoic acid (DHA) and eicosapentaenoic acid (EPA) have a beneficial effect by reducing hepatic inflammation, steatosis and fibrosis in pre-clinical and human models [159]. NAFLD and NASH patients have lower n-3 PUFA levels and diets enriched in n-3 and n-6 PUFA improved the metabolic status [160,161]. DHA and EPA also have a beneficial effect on sarcopenia, as observed in cancer patients and the elderly [162,163]. Nevertheless, a meta-analysis conducted in 2007 did not prove a beneficial effect of EPA in primary sarcopenia [164].

Another cornerstone in the treatment of sarcopenia is physical activity, even though the optimal exercise regimens to prevent and/or reduce sarcopenia in CLD are still debated. Exercise improves the skeletal muscle mass, the physical performance and the muscle strength with a possible reverse of sarcopenia in CLD [165]. A home-based physical activity program, based on a mix of aerobic and resistance mild-intensity exercises, seems to have a favourable impact on muscle mass and strength in cirrhotic patients [166]. In NAFLD patients, a combination of aerobic and resistance exercises reduced steatosis and improved cardiorespiratory fitness [167]. Exercises that improve cardiorespiratory fitness have a beneficial effect on obesity in NAFLD [168], while the effect on sarcopenia is partial. These results have been recently confirmed in a meta-analysis that aimed to assess the impact of training in sarcopenic patients with NAFLD. The authors found that endurance and combined exercises (endurance plus resistance) improved the physical performance evaluated using cardiopulmonary exercise testing but have no effect on muscle mass. However, none of the studies included in the meta-analysis evaluated the impact on muscle strength [169].

### 7.2. Future Directions in the Management of Sarcopenia

Due to the several findings supporting the role of sarcopenia in the early stages of liver disease, several therapeutic approaches are under definition. 

Both NAFLD [170] and sarcopenia [171] feature a dysfunction in the somatotropic axis, although its impact on sarcopenia development in NAFLD patients has not been established yet. The supplementation of growth hormone (GH) and insulin-like growth factor-1 (IGF-1) seems to improve sarcopenia. Indeed, GH and IGF-1 have an anabolic effect with an increase in muscular fibres’ sizes and muscle function [171]. The supplementation with IGF-1 has a higher effect on sarcopenia compared to GH supplementation, probably due to an intrinsic resistance of muscle when chronically exposed to supraphysiologic levels of GH [172]. In NAFLD, low IGF-1 serum levels have been associated with increase liver inflammation and fibrosis, favouring the progression of liver disease [170]. The supplementation of IGF-1 showed a higher reduction of both histological steatosis and sarcopenia, also favoured by a safer metabolic profile compared to GH intake [173]. 

Beta-hydroxy-beta-methylbutyrate (HMB) is a metabolite of leucine which showed benefits for muscle strength and mass in athletes. HMB seems to blunt autophagy and proteolysis in the muscle cells and promote protein synthesis through the activation of mTOR pathways [174] and the increase of GH and IGF-1 levels [175]. Moreover, HMB increases calcium release from the smooth reticulum and the mitochondrial biogenesis, with a better performance during physical exercises, especially in untrained or trained subjects during high physical stress [176]. In sarcopenic old patients, a supplementation of HMB can prevent the loss of lean body mass and exerts potential when combined with physical activity [177]. Despite endogenous HMB being produced in the liver, the use of HMB supplementation in CLD is poorly investigated. 

The imbalance between muscle anabolism and catabolism is the basis of sarcopenia development. The anabolic androgenic steroids are derived from testosterone and showed anabolic and androgenic effects through the binding to androgen receptor. This complex finally stimulates protein synthesis [178]. Oxandrolone is the only anabolic androgenic steroid approved by the Food and Drug Administration (FDA) for the treatment of weight loss after severe trauma, major surgery, alcoholic cirrhosis, Duchenne and Becker muscular dystrophy and severe infections. Oxandrolone increases muscle function and physical performance, increases protein synthesis and intake and reduces visceral fat [179]. The use of oxandrolone is widely limited, especially outside the United States, due to its side effects such as an increase in liver enzymes, androgenic effects, behavioural changes and alterations in cholesterol metabolism. Selective androgen receptor modulators (SARM) exert both antagonist and agonist effects on the androgenic receptors, depending on regulatory proteins in target tissue. SARMs can selectively elicit their anabolic effects, overcoming the androgenic side effects [180]. In fact, it has been shown that SARMs have the capacity to increase muscle mass in rat models, preventing the development of sarcopenia [181]. In humans, enobosarm, a SARM, led to an increase in lean body mass in cancer-related sarcopenia, but it did not improve muscle performance [182]. 

Finally, in murine models, N-acetyl cysteine (NAC) protected against the decrease in muscle strength and mass through the inhibition of caspase-8 and 9 expression, two enzymes involved in cells apoptosis [183].

## 8. Concluding Remarks

Sarcopenia is highly prevalent in CLD of different aetiologies, and it seems to foster the progression of liver disease towards advanced stages and to expose patients to high morbidity and mortality.

Despite most studies being conducted in the advanced stages of liver disease, sarcopenia occurs frequently even in early stages of CLD, independently of the aetiology.

Another feature of muscular impairment in CLD is myosteatosis, although data in non-severe liver diseases are poor, mainly due to the difficulty of an early diagnosis consequent to complex imaging techniques. Nevertheless, as for the data available in the literature, myosteatosis seems to be a common hallmark of NAFLD and its presence is also reported in CHC after viral eradication, probably linked to the presence of concomitant metabolic comorbidities. The co-existence of metabolic comorbidities and myosteatosis worsens liver damage, with a consequent increase in inflammation and fibrosis, as shown in MAFLD patients and NAFLD obese mice. Both sarcopenia and myosteatosis negatively impact the development and progression of NAFLD, ALD and viral hepatitis. Chronic inflammation, IR and mitochondrial dysfunction are the possible underlying pathogenetic mechanisms linking all these clinical entities.

Our review focused on the impact of muscle dysfunction on disease progression in NAFLD, ALD, and viral hepatitis; however, a relationship between sarcopenia and disease severity may also be present in other chronic liver diseases. Few studies are available in autoimmune hepatitis (AIH) and cholestatic liver disease [184,185] and report a relationship between sarcopenia, visceral fat and disease progression, but further studies will be needed to confirm this evidence.

Despite the increased knowledge about sarcopenia, its evaluation in CLD patients is not routine, as demonstrated by the lack of suggestion in International Guidelines [186]. One possible explanation could be the not yet universal definition of sarcopenia, since the use of SMI to define loss in skeletal mass is still debated. In addition, in line with EWGSOP indications, the assessment of sarcopenia is wider, as it should also encompass the evaluation of muscle strength and physical performance beyond muscle mass, but most of the studies evaluated only the latter [1]. Finally, the role of ethnic, gender and genetic background in sarcopenia and CLD remains to be fully investigated.

As sarcopenia plays a crucial role in CLD, treating this condition becomes highly important. Diet and physical activity are the main therapeutic interventions to prevent and treat sarcopenia. Vitamin D and amino acids supplementation, especially with BCAAs, a low sugar diet and the use of antioxidants are recommended. Despite evidence of the benefits of physical activity in sarcopenia, a specific physical activity program tailored to patients with sarcopenia and the early stages of CLD is not defined. Generally, a mix of endurance and resistance exercises showed a beneficial effect, but the majority of studies are conducted in cirrhotic patients without differentiating across aetiologies. Possible future therapeutic approaches have been assessed, mainly in patients with primary sarcopenia or cancer-related sarcopenia, whereas the management of sarcopenia in CLD subjects is still an under-reached field. 

Due to the pathogenetic mechanisms shared by sarcopenia and myosteatosis, the therapeutic approaches are similar for both conditions, as physical activity and a controlled diet seem to improve both of them.

In conclusion, the assessment of sarcopenia in CLD could be helpful in identifying those patients more vulnerable to developing liver disease, and its early management could reduce the progression towards more severe forms. The relationship between sarcopenia and the early stages of CLD should be further investigated in an attempt to identify novel therapeutic approaches.

## Figures and Tables

**Figure 1 biomedicines-10-00182-f001:**
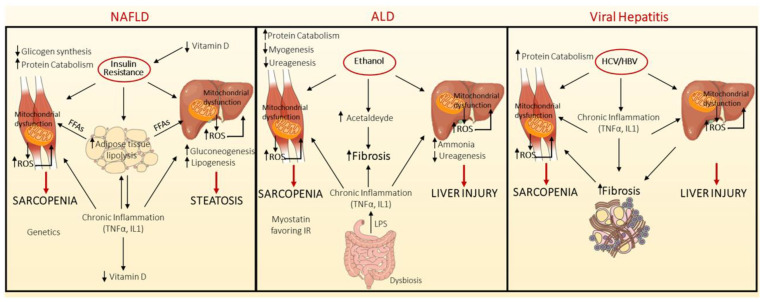
Pathogenic mechanisms linking sarcopenia development in the context of non-alcoholic Fatty Liver Disease (NAFLD), Alcoholic Liver Disease (ALD) and Viral hepatitis. NAFLD and sarcopenia share common underlying pathogenic mechanisms, such as IR, chronic inflammation, mitochondrial dysfunction, nutritional deficiencies, and reduction in physical activity. In the context of NAFLD, the key pathogenic event is the presence of insulin resistance (IR), which is associated with compensatory hyperinsulinemia. Insulin plays a primary role in different tissues among which liver, adipose tissue and muscle. IR in adipose tissue triggers the activation of lipolysis, favouring in turn the release of free fatty acids (FFAs) into the bloodstream. At the hepatic level, the reduced insulin signalling hampers the phosphorylation of Forkhead box O1-phosphorylated (FOXO1), whereby forcing gluconeogenesis, while the enhanced influx of FFAs induces fat deposition, endoplasmic reticulum (ER) stress, mitochondrial dysfunction, and reactive oxygen species (ROS) production. Moreover, hyperinsulinemia stimulates de novo lipogenesis, through sterol regulatory element-binding protein-1c (SREBP1c). In muscle tissue, IR suppresses glycogen synthesis and muscle mass enlargement. In addition, both sarcopenia and NAFLD are shaped by a chronic inflammation, characterized by tumour necrosis factor alpha (TNFα) and interleukin 1 (IL1) over-secretion. Similarly, Vitamin D deficiency participates to sarcopenia onset. The gut microbiota and the somatotropic axis are involved both in sarcopenia and NAFLD, but their effect on sarcopenia development and progression in NAFLD is not well established (**left** panel). Likewise, in Alcoholic Liver disease (ALD), ethanol exerts a detrimental effect on both muscle and liver, through its conversion into acetaldehyde, which induces ROS production, mitochondrial dysfunction, promotes the activation of hepatic stellate cells (HSCs) producing extracellular matrix (ECM) deposition, and interferes with hepatic ureagenesis with consequent increase in ammonia levels. Hyperammonaemia per se blunted protein synthesis in skeletal muscle and stimulates myostatin, a member of transforming growth factor beta (TGFβ) (**middle** panel). Less is known regarding the mechanisms linking chronic viral hepatitis and sarcopenia. In this background, chronic inflammation promotes the protein catabolism in muscle cells, by ubiquitin-proteasome system (UPS) (**right** panel).

## Data Availability

Not applicable.

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
