# Peer review of "Impact of Sarcopenia and Myosteatosis in Non-Cirrhotic Stages of Liver Diseases: Similarities and Differences across Aetiologies and Possible Therapeutic Strategies"

_biomedicines, 2022, doi:10.3390/biomedicines10010182_

Round 1
Reviewer 1 Report
The manuscript “Impact of Sarcopenia and Myosteatosis in non-cirrhotic stages of liver diseases: similarities and differences across aetiologies and possible therapeutic strategies” describes the correlation between chronic liver diseases including non-alcoholic steatohepatitis (NASH), alcoholic liver disease (ALD), hepatitis B &C and Sarcopenia and Myosteatosis.
- This is a very well planned, written and well researched review which will add the knowledge to the field.
- Few spelling mistakes were observed in review such as “glycolysis” is written as “glycolysis” and “dysfunction” is written as “disfunction”. Please read the manuscript thoroughly and correct those mistakes.
Author Response
We sincerely thank the Reviewer for her/his thoughtful revision of the manuscript and for her/his suggestions.
Please see below the point-to-point answers to the Reviewer. Changes in the text are marked up using the “Track Changes” (which are in red in the text) and highlighted in yellow.
The manuscript “Impact of Sarcopenia and Myosteatosis in non-cirrhotic stages of liver diseases: similarities and differences across aetiologies and possible therapeutic strategies” describes the correlation between chronic liver diseases including non-alcoholic steatohepatitis (NASH), alcoholic liver disease (ALD), hepatitis B &C and Sarcopenia and Myosteatosis.
Point 1: This is a very well planned, written and well researched review which will add the knowledge to the field.
Response 1: We really thank the Reviewer for the positive comments on the manuscript.
Point 2: Few spelling mistakes were observed in review such as “glycolysis” is written as “glycolysis” and “dysfunction” is written as “disfunction”. Please read the manuscript thoroughly and correct those mistakes.
Response 2: We apologize for the typos. We have now checked the entire manuscript and corrected the mistakes.
Reviewer 2 Report
In this manuscript titled “Impact of Sarcopenia and Myosteatosis in non-cirrhotic stages of liver diseases: similarities and differences across aetiologies and possible therapeutic strategies”, the authors summarized the recent discoveries, especially clinical findings, about the relationship between Sarcopenia, Myosteatosis and liver diseases, and further introduced the treatment on Sarcopenia. Overall, this is a relatively comprehensive and systemic review, which summarized the recent advancement and puzzles in the field very well. It is well written and easy to understand. A revision on the following minor issues may further improve the quality of this manuscript.
Minor issues:
- The authors didn’t spend much effort on the discussion of myosteatosis. Specifically, the authors didn’t introduce the treatment on myosteatosis, and the authors didn’t summarize the relationship between myosteatosis and liver diseases in the concluding remarks.
- In the abstract, “… in the early stage of CLD across different aetiologies (mainly non-alcoholic fatty liver disease, alcohol and viral hepatitis)”, please change “alcohol” into “alcohol-related liver disease”.
- In the introduction “To date, the prevalence of sarcopenia ranges from 2% to 25%, according to the diagnostic tools”, it is not clear what the “2% to 25%” means. Does it mean the percentage among total population in a certain country, the percentage among a population with a certain age, or the percentage among a population with a certain medical condition?
- In the section “2. Sarcopenia in NAFLD and Metabolic Associated Comorbidities”, “Afterwards, an overwhelming number of studies confirmed this independent association [30, 31]”. The usage of “independent” is a little confusing here. The author seems to refer to an association independent of other risk factors, but “independent association” is a weird phrase. It may be better to delete “independent” in this sentence.
- In the section “2. Sarcopenia in NAFLD and Metabolic Associated Comorbidities”, “Therefore, to date, it remains unclear whether NAFLD is a consequence or rather a cause of sarcopenia”. It is very hard to establish causation in clinical studies because it is very hard to do experiments on human. Could the authors comment on whether there are animal models available to study the causation relationship between NAFLD and sarcopenia? Or is there any study on SMM in animal models of NAFLD? If not, could the authors comment on whether studies on animal models will help the establishment of this causation relationship?
- In the section “2. Sarcopenia in NAFLD and Metabolic Associated Comorbidities”, “Finally, the role of the ethnic background, gender and genetic predisposition has not been defined yet. Most of the studies have been performed in in Asian subjects whereas those performed in Caucasians described an association between sarcopenia and NAFLD mainly in women”. Please elaborate on this point. First, please note that Asians and Caucasians are not the only ethnic groups in the world, please tough upon other ethnic groups too. Second, the authors didn’t give the details about genetic predisposition in this sentence.
- The formatting of numbers in this manuscript is a little weird. For example, in the section “3. Sarcopenia in Alcohol-related Liver Disease”, “as shown in a meta-analysis which included 13 studies thus encompassing 13.155 subjects”, 13.155 should be 13,155. There are several places in this manuscript like this one, please double check the formatting throughout this manuscript.
- In the section “4.1. Prevalence of Sarcopenia in Viral Hepatitis”, “Sarcopenia is also observed in patients affected by chronic hepatitis B (CHB) and C (CHC) infection”. Please delete “infection”.
- In the section “6.1. Future directions in the management of Sarcopenia”, IGF-1 was mentioned for multiple times. However, the authors didn’t mention if IGF-1 play any roles in sarcopenia and liver diseases, and the authors didn’t mention if the level of IGF-1 changes in patients with liver diseases. Is it a pure empirical finding that administration of IGF-1 can manage sarcopenia or is there any indicator in the patients with liver diseases showing that manipulating IGF-1 could be critical?
Author Response
We sincerely thank the Reviewer for her/his thoughtful revision of the manuscript and for her/his suggestions.
Please see below the point-to-point answers to the Reviewer. Changes in the text are marked up using the “Track Changes” (which are in red in the text) and highlighted in yellow.
In this manuscript titled “Impact of Sarcopenia and Myosteatosis in non-cirrhotic stages of liver diseases: similarities and differences across aetiologies and possible therapeutic strategies”, the authors summarized the recent discoveries, especially clinical findings, about the relationship between Sarcopenia, Myosteatosis and liver diseases, and further introduced the treatment on Sarcopenia. Overall, this is a relatively comprehensive and systemic review, which summarized the recent advancement and puzzles in the field very well. It is well written and easy to understand. A revision on the following minor issues may further improve the quality of this manuscript.
Point 1: The authors didn’t spend much effort on the discussion of myosteatosis. Specifically, the authors didn’t introduce the treatment on myosteatosis, and the authors didn’t summarize the relationship between myosteatosis and liver diseases in the concluding remarks.
Response 1: We really thank the Reviewer for her/his interesting comment. In this review we mainly considered the current and possible therapeutic approaches to address sarcopenia. Nevertheless, as suggested by the Reviewer we added a paragraph concerning the possible treatment of myosteatosis and we also better summarize the relationship between myosteatosis and CLD.
Point 2: In the abstract, “… in the early stage of CLD across different aetiologies (mainly non-alcoholic fatty liver disease, alcohol and viral hepatitis)”, please change “alcohol” into “alcohol-related liver disease”.
Response 2: As suggested by the Reviewer, we have changed the sentence.
Point 3: In the introduction “To date, the prevalence of sarcopenia ranges from 2% to 25%, according to the diagnostic tools”, it is not clear what the “2% to 25%” means. Does it mean the percentage among total population in a certain country, the percentage among a population with a certain age, or the percentage among a population with a certain medical condition?
Response 3: We thank the Reviewer for her/his comment. The prevalence of sarcopenia ranges from 2% to 25% in people aged 50 years old and over. We have clarified it in the manuscript.
Point 4: In the section “2. Sarcopenia in NAFLD and Metabolic Associated Comorbidities”, “Afterwards, an overwhelming number of studies confirmed this independent association [30, 31]”. The usage of “independent” is a little confusing here. The author seems to refer to an association independent of other risk factors, but “independent association” is a weird phrase. It may be better to delete “independent” in this sentence.
Response 4: We agree with the Reviewer, and we have deleted the word “independent”.
Point 5: In the section “2. Sarcopenia in NAFLD and Metabolic Associated Comorbidities”, “Therefore, to date, it remains unclear whether NAFLD is a consequence or rather a cause of sarcopenia”. It is very hard to establish causation in clinical studies because it is very hard to do experiments on human. Could the authors comment on whether there are animal models available to study the causation relationship between NAFLD and sarcopenia? Or is there any study on SMM in animal models of NAFLD? If not, could the authors comment on whether studies on animal models will help the establishment of this causation relationship?
Response 5: We completely agree with the Reviewer. It is difficult to assess the causal relationship between sarcopenia and NAFLD because the lack of longitudinal studies in humans and the few pre-clinical models. We added the only study, to our knowledge, available in literature concerning sarcopenia in an animal model of NAFLD.
Point 6: In the section “2. Sarcopenia in NAFLD and Metabolic Associated Comorbidities”, “Finally, the role of the ethnic background, gender and genetic predisposition has not been defined yet. Most of the studies have been performed in in Asian subjects whereas those performed in Caucasians described an association between sarcopenia and NAFLD mainly in women”. Please elaborate on this point. First, please note that Asians and Caucasians are not the only ethnic groups in the world, please tough upon other ethnic groups too. Second, the authors didn’t give the details about genetic predisposition in this sentence.
Response 6: We thank the Reviewer for pointing this out. Despite there are several studies about sarcopenia across several ethnic group, there are few studies concerning the relationship between sarcopenia and early stages of CLD in non-Asian and non-Caucasian subjects, major conducted in children. We have better discussed it into manuscript. We also better detailed the findings concerning genetic predisposition.
Point 7: The formatting of numbers in this manuscript is a little weird. For example, in the section “3. Sarcopenia in Alcohol-related Liver Disease”, “as shown in a meta-analysis which included 13 studies thus encompassing 13.155 subjects”, 13.155 should be 13,155. There are several places in this manuscript like this one, please double check the formatting throughout this manuscript.
Response 7: We apologize for the mistakes in numbers format. We corrected them in the manuscript.
Point 8: In the section “4.1. Prevalence of Sarcopenia in Viral Hepatitis”, “Sarcopenia is also observed in patients affected by chronic hepatitis B (CHB) and C (CHC) infection”. Please delete “infection”.
Response 8: We thank the Reviewer for this observation. We deleted the term “infection”.
Point 9: In the section “6.1. Future directions in the management of Sarcopenia”, IGF-1 was mentioned for multiple times. However, the authors didn’t mention if IGF-1 play any roles in sarcopenia and liver diseases, and the authors didn’t mention if the level of IGF-1 changes in patients with liver diseases. Is it a pure empirical finding that administration of IGF-1 can manage sarcopenia or is there any indicator in the patients with liver diseases showing that manipulating IGF-1 could be critical?
Response 9: We really thank the Reviewer for these observations and we completely agree. Several studies showed a dysfunction of the GH/IGF-1 axis both in sarcopenia and NAFLD but its role in the development of sarcopenia in NAFLD patients is not well defined. We have better specified the role of supplementation of IGF-1 in liver disease.